# Improved Estimation of Concentration Under $\ell_p$-Norm Distance Metrics Using Half Spaces

**Jack B. Prescott, Xiao Zhang and David Evans**
Department of Computer Science
University of Virginia
{jbp2jn, shawn, evans}@virginia.edu

## Abstract

Concentration of measure has been argued to be the fundamental cause of adversarial vulnerability. Mahloujifar et al. (2019b) presented an empirical way to measure the concentration of a data distribution using samples, and employed it to find lower bounds on intrinsic robustness for several benchmark datasets. However, it remains unclear whether these lower bounds are tight enough to provide a useful approximation for the intrinsic robustness of a dataset. To gain a deeper understanding of the concentration of measure phenomenon, we first extend the Gaussian Isoperimetric Inequality to non-spherical Gaussian measures and arbitrary $\ell_p$-norms ($p \geq 2$). We leverage these theoretical insights to design a method that uses half-spaces to estimate the concentration of any empirical dataset under $\ell_p$-norm distance metrics. Our proposed algorithm is more efficient than Mahloujifar et al. (2019b)'s, and our experiments on synthetic datasets and image benchmarks demonstrate that it is able to find much tighter intrinsic robustness bounds. These tighter estimates provide further evidence that rules out intrinsic dataset concentration as a possible explanation for the adversarial vulnerability of state-of-the-art classifiers.

## 1 Introduction

Despite achieving exceptional performance in benign settings, modern machine learning models have been shown to be highly vulnerable to inputs, known as *adversarial examples*, crafted with targeted but imperceptible perturbations (Szegedy et al., 2014; Goodfellow et al., 2015). This discovery has prompted a wave of research studies to propose defense mechanisms, including heuristic approaches (Papernot et al., 2016; Mądry et al., 2018; Zhang et al., 2019) and certifiable methods (Wong & Kolter, 2018; Gowal et al., 2019; Cohen et al., 2019). Unfortunately, none of these methods can successfully produce adversarially-robust models, even for classification tasks on toy datasets such as CIFAR-10. To explain the prevalence of adversarial examples, a line of theoretical works (Gilmer et al., 2018; Fawzi et al., 2018; Shafahi et al., 2019; Dohmatob, 2019; Bhagoji et al., 2019) have proven upper bounds on the maximum achievable adversarial robustness by imposing different assumptions on the underlying metric probability space. In particular, Mahloujifar et al. (2019a) generalized the previous results showing that adversarial examples are inevitable as long as the input distributions are concentrated with respect to the perturbation metric. Thus, the question of whether or not natural image distributions are concentrated is highly relevant, as if they are it would rule out any possibility of there being adversarially robust image classifiers.

Recently, Mahloujifar et al. (2019b) proposed an empirical method to measure the concentration of an arbitrary distribution using data samples, then employed it to estimate a lower bound on *intrinsic robustness* (see Definition 2.2 for its formal definition) for several image benchmarks. By demonstrating the gap between the estimated bounds of intrinsic robustness and the robustness performance achieved by the best current models, they further concluded concentration of measure is not the sole reason behind the adversarial vulnerability of existing classifiers for benchmark image distributions. However, due to the heuristic nature of the proposed algorithm, it remains elusive whether the estimates it produces can serve as useful approximations of the underlying intrinsic robustness limits, thus hindering understanding of how much of the actual adversarial risk can be explained by the concentration of measure phenomenon.

In this work, we address this issue by first characterizing the optimum of the actual concentration problem for general Gaussian spaces, then using our theoretical insights to develop an alternative algorithm for measuring concentration empirically that significantly improves both the accuracy and efficiency of estimates of intrinsic robustness. While we do not demonstrate a specific classifier which achieves this robustness upper bound, our results rule out inherent image distribution concentration as the reason for our current inability to find adversarially robust models.

**Contributions.** We generalize the Gaussian Isoperimetric Inequality to non-spherical Gaussian distributions and $\ell_p$-norm distance metrics with $p \geq 2$ (including $\ell_\infty$) (Theorem 3.3). Motivated by the optimal concentration results for special Gaussian spaces (Remark 3.4), we develop a sample-based algorithm to estimate the concentration of measure using half spaces that works for arbitrary distribution and any $\ell_p$-norm distance (Section 4). Compared with prior approaches, we empirically demonstrate the significant increase in efficacy of our method under $\ell_\infty$-norm distance metric (Section 6). Not only does the proposed method converge to its limit with an order of magnitude fewer data (Section 6.2), it also finds a much tighter lower bound of intrinsic robustness for both simulated datasets whose underlying concentration function is analytically derivable and various benchmark image datasets (Section 6.1). In particular, we improve the best current estimated lower bound of intrinsic robustness from approximately $82\%$ to above $93\%$ for CIFAR-10 under $\ell_\infty$-norm bounded perturbations with $\epsilon = 8/255$. These tighter concentration estimates produced by our algorithm provide strong evidence that concentration of measure should not be considered as the main cause of adversarial vulnerability, at least for the image benchmarks evaluated in our experiments.

**Related Work.** Several prior works have sought to empirically estimate lower bounds on intrinsic robustness using data samples. The pioneering work of Gilmer et al. (2018) introduced the connection between adversarial examples and the concentration phenomenon for uniform $n$-spheres, then proposed a simple heuristic to find a half space that expands slowly under Euclidean distance for the MNIST dataset. Our work can be seen as a strict generalization of Gilmer et al. (2018)'s, which applies to arbitrary $\ell_p$-norm distance metrics (including $\ell_\infty$). By characterizing the optimal transport cost between conditional distributions, Bhagoji et al. (2019) estimated a lower bound on the best possible adversarial robustness for several image datasets. However, when applied to adversaries beyond $\ell_2$, such as $\ell_\infty$, the lower bound produced by their method is not informative (that is, it is close to zero). The most relevant previous work is Mahloujifar et al. (2019b), which proposed a general method for measuring concentration using special collections of subsets. Although the optimal value of the considered empirical concentration problem is proven to asymptotically converge to the actual concentration, there is no guarantee that the proposed searching algorithm for solving the empirical problem finds the optimum. Our approach follows the framework introduced by Mahloujifar et al. (2019b)'s, but considers a different collection of subsets for the empirical concentration problem. This not only results in optimality for theoretical Gaussian distributions, but also significantly improves the estimation performance for typical image benchmarks.

Another line of work attempts to provide estimates of intrinsic robustness upper bounds based on generative assumptions. In order to justify the theoretically-derived impossibility results, Fawzi et al. (2018) estimated the smoothness parameters of the state-of-the-art generative models on CIFAR-10 and SVHN datasets, which yield approximated upper bounds on adversarial robustness for any classifiers. Zhang et al. (2020) generalized their results to non-smoothed data manifolds, such as datasets that can be captured by a conditional generative model. However, these methods only work for simulated generative distributions, which may deviate from the actual distributions they are intended to understand.

**Notation.** For any $n \in \mathbb{Z}^+$, denote by $[n]$ the set $\{1, 2, \ldots, n\}$. Lowercase boldface letters denote vectors and uppercase boldface letters represent matrices. For any vector $\boldsymbol{x}$ and $p \in [1, \infty)$, let $x_j$, $\|\boldsymbol{x}\|_p$ and $\|\boldsymbol{x}\|_\infty$ be the $j$-th element, the $\ell_p$-norm and the $\ell_\infty$-norm of $\boldsymbol{x}$. For any matrix $\mathbf{A}$, $\mathbf{B}$ is said to be a square root of $\mathbf{A}$ if $\mathbf{A} = \mathbf{BB}$, and the induced matrix $p$-norm of $\mathbf{A}$ is defined as $\|\mathbf{A}\|_p = \sup_{\boldsymbol{x} \neq \mathbf{0}} \{\|\mathbf{A}\boldsymbol{x}\|_p / \|\boldsymbol{x}\|_p\}$. Denote by $\mathcal{N}(\boldsymbol{\theta}, \boldsymbol{\Sigma})$ the Gaussian distribution with mean $\boldsymbol{\theta}$ and covariance matrix $\boldsymbol{\Sigma}$. Let $\gamma_n$ be the probability measure of $\mathcal{N}(\mathbf{0}, \mathbf{I}_n)$, where $\mathbf{I}_n$ denotes the identity matrix. Let $\Phi(\cdot)$ be the cumulative distribution function of $\mathcal{N}(0, 1)$ and $\Phi^{-1}(\cdot)$ be its inverse. For any set $\mathcal{A}$, let $\text{pow}(\mathcal{A})$ and $\mathbb{1}_{\mathcal{A}}(\cdot)$ be all measurable subsets and the indicator function of $\mathcal{A}$. Let $(\mathcal{X}, \mu, \Delta)$ be a metric probability space, where $\Delta : \mathcal{X} \times \mathcal{X} \to \mathbb{R}_{\geq 0}$ denotes a distance metric on $\mathcal{X}$. Define the empirical measure with respect to a sample set $\{\boldsymbol{x}_i\}_{i \in [m]}$ as $\widehat{\mu}_m(\mathcal{A}) = \frac{1}{m} \sum_{i \in [m]} \mathbb{1}_{\mathcal{A}}(\boldsymbol{x}_i)$,

$\forall \mathcal{A} \in \mathsf{pow}(\mathcal{X})$. Let $\mathcal{B}(\boldsymbol{x}, \epsilon, \Delta) = \{\boldsymbol{x}' \in \mathcal{X} : \Delta(\boldsymbol{x}', \boldsymbol{x}) \leq \epsilon\}$ be the ball around $\boldsymbol{x}$ with $\epsilon$ radius. Define the $\epsilon$-expansion of $\mathcal{A}$ as $\mathcal{A}_\epsilon^{(\Delta)} = \{\boldsymbol{x} \in \mathcal{X} : \exists \, \boldsymbol{x}' \in \mathcal{B}(\boldsymbol{x}, \epsilon, \Delta) \cap \mathcal{A}\}$.

## 2 PRELIMINARIES

In this section, we introduce the problem of measuring concentration and its connection to adversarial robustness. Consider a metric probability space of instances $(\mathcal{X}, \mu, \Delta)$. Given parameters $\epsilon \geq 0$ and $\alpha > 0$, the concentration of measure problem[1] can be cast as the following optimization problem:

$$\underset{\mathcal{E} \in \mathsf{Pow}(\mathcal{X})}{\text{minimize}} \; \mu\big(\mathcal{E}_\epsilon^{(\Delta)}\big) \quad \text{subject to} \; \mu(\mathcal{E}) \geq \alpha. \tag{2.1}$$

We focus on the case where $\Delta$ is some $\ell_p$-norm distance metric (including $\ell_\infty$) in this work.

Concentration of measure has been shown to be closely related to adversarial examples (Gilmer et al., 2018; Fawzi et al., 2018; Mahloujifar et al., 2019a). In particular, one can prove that for a given robust learning problem, if the input distribution is concentrated with respect to the perturbation metric, no adversarially robust model exists. The concentration parameter (which corresponds to the optimal value of optimization problem (2.1)) determines an inherent upper bound on the maximum adversarial robustness that *any* model can achieve for the given problem.

To explain the connection between concentration of measure and robust learning in a more formal way, we lay out the definition of *adversarial risk* that we work with. We draw this definition from several previous works, including Gilmer et al. (2018); Bubeck et al. (2019); Mahloujifar et al. (2019a;b).[2]

**Definition 2.1** (Adversarial Risk). Let $(\mathcal{X}, \mu, \Delta)$ be the input metric probability space. Assume $f^*$ is the underlying ground-truth classifier that gives labels to any input. Given classifier $f$ and $\epsilon \geq 0$, the *adversarial risk* of $f$ with respect to $\epsilon$-perturbations measured by $\Delta$ is defined as:

$$\text{AdvRisk}_\epsilon(f) = \Pr_{\boldsymbol{x} \sim \mu} \big[ \exists \, \boldsymbol{x}' \in \mathcal{B}(\boldsymbol{x}, \epsilon, \Delta) \text{ s.t. } f(\boldsymbol{x}') \neq f^*(\boldsymbol{x}') \big].$$

Correspondingly, we define the *adversarial robustness* of $f$ as $\text{AdvRob}_\epsilon(f) = 1 - \text{AdvRisk}_\epsilon(f)$.

When $\epsilon = 0$, adversarial risk degenerates to standard risk. In other words, it holds for any $f$ that $\text{AdvRisk}_0(f) = \text{Risk}(f) := \Pr_{\boldsymbol{x} \sim \mu}[f(\boldsymbol{x}) \neq f^*(\boldsymbol{x})]$. We remark that this definition assumes the existence of an underlying ground-truth labeling function, which does not apply to the agnostic setting where inputs can have non-deterministic labels.

Initially introduced in Mahloujifar et al. (2019b), intrinsic robustness captures the maximum adversarial robustness that can be achieved by any imperfect classifier for a robust classification problem.

**Definition 2.2** (Intrinsic Robustness). Consider the same setting as in Definition 2.1. For any $\alpha > 0$, let $\mathcal{F}_\alpha = \{f : \text{Risk}(f) \geq \alpha\}$ be the set of imperfect classifiers whose risk is at least $\alpha$. Then the *intrinsic robustness* of the given robust classification problem with respect to $\mathcal{F}_\alpha$ is defined as:

$$\overline{\text{AdvRob}}_\epsilon(\mathcal{F}_\alpha) = 1 - \inf_{f \in \mathcal{F}_\alpha} \big\{ \text{AdvRisk}_\epsilon(f) \big\} = \sup_{f \in \mathcal{F}_\alpha} \big\{ \text{AdvRob}_\epsilon(f) \big\}.$$

It is worth noting that the value of $\overline{\text{AdvRob}}_\epsilon(\mathcal{F}_\alpha)$ is only determined by the underlying input data distribution, the perturbation set and the risk threshold parameter $\alpha$, which is independent of the model class one would choose for learning.

By relating the robustness of a classifier to the $\epsilon$-expansion of its induced error region, the following lemma, proved in Mahloujifar et al. (2019a), establishes a fundamental connection between the concentration of measure and the intrinsic robustness one can hope for a robust classification problem.

**Lemma 2.3.** Consider the same setting as in Definition 2.2. Let $h_\mu^{(\Delta)}(\alpha, \epsilon)$ be the concentration function that captures the optimal value of the concentration of measure problem (2.1):

$$h_\mu^{(\Delta)}(\alpha, \epsilon) = \inf \big\{ \mu\big(\mathcal{E}_\epsilon^{(\Delta)}\big) : \mathcal{E} \in \mathsf{pow}(\mathcal{X}) \text{ and } \mu(\mathcal{E}) \geq \alpha \big\}.$$

Then, $\overline{\text{AdvRob}}_\epsilon(\mathcal{F}_\alpha) = 1 - h_\mu^{(\Delta)}(\alpha, \epsilon)$ holds for any $\alpha > 0$ and $\epsilon \geq 0$.

---

[1] The standard notion of concentration of measure (Talagrand, 1995) corresponds to the case where $\alpha = 0.5$.

[2] Other related definitions, such as the one used in most empirical works for robustness evaluation, are equivalent to this, as long as small perturbations preserve the ground truth. See Diochnos et al. (2018) for a detailed comparison of these and other definitions of adversarial robustness.

Lemma 2.3 suggests that one can characterize the intrinsic robustness limit for a robust classification problem by measuring the concentration of the input data with respect to the perturbation metric.

In this paper, we aim to understand and empirically estimate the intrinsic robustness limit for typical robust classification tasks by measuring concentration. It is worth noting that solving the concentration problem (2.1) itself only shows the existence of an error region $\mathcal{E}$ whose $\epsilon$-expansion has certain (small) measure. This further implies the possibility of existing an optimally robust classifier (with risk at least $\alpha$), whose robustness matches the intrinsic robustness limit $\overline{\mathrm{AdvRob}}_\epsilon(\mathcal{F}_\alpha)$. However, actually finding such optimal classifier using a learning algorithm might be a much more challenging task, which is beyond the scope of this work.

## 3 GENERALIZING THE GAUSSIAN ISOPERIMETRIC INEQUALITY

Before proceeding to introduce the proposed methodology for solving the concentration of measure problem, we first present our main theoretical results of generalizing the Gaussian Isoperimetric Inequality. This theoretical result largely motivates our method.

To begin with, we introduce the Gaussian Ispoperimetric Inequality (Sudakov & Tsirelson, 1974; Borell, 1975). It characterizes the optimum of the concentration problem (2.1) with respect to standard Gaussian distribution and $\ell_2$-distance, where half spaces are proven to be the optimal sets.

**Definition 3.1** (Half Space). Let $\boldsymbol{w} \in \mathbb{R}^n$ and $b \in \mathbb{R}$. Without loss of generality, assume $\|\boldsymbol{w}\|_2 = 1$. An $n$-dimensional *half space* with parameters $\boldsymbol{w}$ and $b$ is defined as:

$$\mathcal{H}_{\boldsymbol{w},b} = \{\boldsymbol{z} \in \mathbb{R}^n : \boldsymbol{w}^\top \boldsymbol{z} + b \leq 0\}.$$

**Lemma 3.2** (Gaussian Isoperimetric Inequality). Consider the standard Gaussian space $(\mathbb{R}^n, \gamma_n)$ with $\ell_2$-distance. Let $\mathcal{E} \in \mathrm{pow}(\mathbb{R}^n)$ and $\mathcal{H}$ be a half space such that $\gamma_n(\mathcal{E}) = \gamma_n(\mathcal{H})$, then for any $\epsilon \geq 0$, it holds that

$$\gamma_n\big(\mathcal{E}_\epsilon^{(\ell_2)}\big) \geq \gamma_n\big(\mathcal{H}_\epsilon^{(\ell_2)}\big) = \Phi\big(\Phi^{-1}\big(\gamma_n(\mathcal{E})\big) + \epsilon\big).$$

The proof of the Gaussian Isoperimetric Inequality can be found in Ledoux (1996).

Lemma 3.2 implies the concentration function of $(\mathbb{R}^n, \gamma_n, \|\cdot\|_2)$ is $h_{\gamma_n}^{(\ell_2)}(\alpha, \epsilon) = \Phi\big(\Phi^{-1}(\alpha) + \epsilon\big)$, but only applies when the underlying distribution is a spherical Gaussian and the metric function is $\ell_2$-distance. Thus, it only gives a concentration function for estimating the the intrinsic robustness limit in a very restrictive setting. To understand the concentration of measure for more general problems, we prove the following theorem that extends the standard Gaussian Isoperimetric Inequality (Lemma 3.2) to non-spherical Gaussian measure and general $\ell_p$-norm distance metrics for any $p \geq 2$.

**Theorem 3.3** (Generalized Gaussian Isoperimetric Inequality). Let $\nu$ be the probability measure of $\mathcal{N}(\boldsymbol{\theta}, \boldsymbol{\Sigma})$, where $\boldsymbol{\theta} \in \mathbb{R}^n$ and $\boldsymbol{\Sigma}$ is a positive definite matrix in $\mathbb{R}^{n \times n}$. Consider the probability space $(\mathbb{R}^n, \nu)$ with $\ell_p$-norm distance, where $p \geq 2$ (including $\ell_\infty$). For any $\mathcal{E} \in \mathrm{pow}(\mathbb{R}^n)$ and $\epsilon \geq 0$,

$$\nu\big(\mathcal{E}_\epsilon^{(\ell_p)}\big) \geq \Phi\big(\Phi^{-1}\big(\nu(\mathcal{E})\big) + \epsilon/\|\boldsymbol{\Sigma}^{1/2}\|_p\big), \tag{3.1}$$

where $\boldsymbol{\Sigma}^{1/2}$ is the square root of $\boldsymbol{\Sigma}$, and $\|\boldsymbol{\Sigma}^{1/2}\|_p$ denotes the induced matrix $p$-norm of $\boldsymbol{\Sigma}^{1/2}$.

**Remark 3.4.** Theorem 3.3 suggests that for general Gaussian distribution $\mathcal{N}(\boldsymbol{\theta}, \boldsymbol{\Sigma})$ and any $\ell_p$-norm distance ($p \geq 2$), the corresponding concentration function is lower bounded by $\Phi(\Phi^{-1}(\alpha) + \epsilon/\|\boldsymbol{\Sigma}^{1/2}\|_p)$. Due to the NP-hardness of approximating the matrix $p$-norm (Hendrickx & Olshevsky, 2010), it is generally hard to infer whether the equality of (3.1) can be attained or not. However, for specific special Gaussian spaces, we can derive optimal subsets that achieve the lower bound. In particular, for the case where $\boldsymbol{\Sigma} = \mathbf{I}_n$ and $p > 2$, the optimum is attained when $\mathcal{E}$ is a half space with axis-aligned weight vector (that is, $\boldsymbol{w} = \mathbf{e}_j$ for some $j \in [n]$). For the case where $\boldsymbol{\Sigma} \neq \mathbf{I}_n$ and $p = 2$, the optimal solution is a half space $\mathcal{H}_{\boldsymbol{v}_1, b}$, where $\boldsymbol{v}_1$ is the eigenvector with respect to the largest eigenvalue of $\boldsymbol{\Sigma}$. The proofs of these optimality results are provided in Appendix A.2.

## 4 EMPIRICALLY MEASURING CONCENTRATION USING HALF SPACES

Recall that the primary goal of this paper is to measure the concentration of an arbitrary distribution. However, for typical classification problems, we might not know the density function of the underlying

distribution $\mu$, but instead we usually have access to a finite set of $m$ instances $\{\boldsymbol{x}_i\}_{i \in [m]}$ sampled from $\mu$. Following Mahloujifar et al. (2019b), we consider the following empirical counterpart of the actual concentration problem (2.1):

$$\underset{\mathcal{E} \in \mathcal{G}}{\text{minimize}} \; \widehat{\mu}_m\big(\mathcal{E}_\epsilon^{(\Delta)}\big) \quad \text{subject to} \; \widehat{\mu}_m(\mathcal{E}) \geq \alpha, \tag{4.1}$$

where $\widehat{\mu}_m$ is the empirical measure based on $\{\boldsymbol{x}_i\}_{i \in [m]}$ and $\mathcal{G} \subseteq \text{pow}(\mathcal{X})$ denotes a particular collection of subsets. Mahloujifar et al. (2019b) proposed the complement of union of $T$ hyperrectangles as $\mathcal{G}$ for $\ell_\infty$ and the union of $T$ balls for $\ell_2$. They proved that if one increases the complexity parameter $T$ and the sample size $m$ together in a careful way, the optimal value of the empirical concentration problem (4.1) converges to the actual concentration asymptotically. However, it is unclear how quickly it converges and how well the proposed heuristic algorithm in Mahloujifar et al. (2019b) finds the optimum of (4.1).

In this work, we argue that the set of half spaces is a superior choice for $\mathcal{G}$ with respect to any $\ell_p$-norm distance. Apart from achieving the optimality for certain Gaussian spaces as discussed in Remark 3.4, estimating concentration using half spaces has several other advantages including the closed-form solution of $\ell_p$-distance to half-space (Lemma 4.1) and its small sample complexity requirement for generalization (Theorem 4.2). To be more specific, we focus on the following optimization problem based on the empirical measure $\widehat{\mu}_m$ and the collection of half spaces $\mathcal{HS}(n)$:

$$\underset{\mathcal{E} \in \mathcal{HS}(n)}{\text{minimize}} \; \widehat{\mu}_m\big(\mathcal{E}_\epsilon^{(\ell_p)}\big) \quad \text{subject to} \; \widehat{\mu}_m(\mathcal{E}) \geq \alpha, \tag{4.2}$$

where $\mathcal{HS}(n) = \{\mathcal{H}_{\boldsymbol{w},b} : \boldsymbol{w} \in \mathbb{R}^n, b \in \mathbb{R}, \text{ and } \|\boldsymbol{w}\|_2 = 1\}$ is the set of all half spaces in $\mathbb{R}^n$.

The following lemma, proven in Appendix B.1, characterizes the closed-form solution of the $\ell_p$-norm distance between a point $\boldsymbol{x}$ and a half space. Such a formulation enables an exact computation of the empirical measure with respect to the $\epsilon$-expansion of any half space.

**Lemma 4.1** ($\ell_p$-Distance to Half Space). Let $\mathcal{H}_{\boldsymbol{w},b} \in \mathcal{HS}(n)$ be an $n$-dimensional half space. For any vector $\boldsymbol{x} \in \mathbb{R}^n$, the $\ell_p$-norm distance ($p \geq 1$) from $\boldsymbol{x}$ to $\mathcal{H}_{\boldsymbol{w},b}$ is:

$$d_p(\boldsymbol{x}, \mathcal{H}_{\boldsymbol{w},b}) = \begin{cases} 0, & \boldsymbol{w}^\top \boldsymbol{x} + b \leq 0; \\ (\boldsymbol{w}^\top \boldsymbol{x} + b)/\|\boldsymbol{w}\|_q, & \text{otherwise.} \end{cases}$$

Here, $q$ is a real number that satisfies $1/p + 1/q = 1$.

Lemma 4.1 implies that the $\epsilon$-expansion of any half space with respect to the $\ell_p$-norm is still a half space. Since the VC-dimensions of both the set of half spaces and its expansion are bounded, we can thus apply the generalization theorem in Mahloujifar et al. (2019b), which yields the following theorem, proved in Appendix B.2, that characterizes the generalization of concentration with respect to the collection of half spaces.

**Theorem 4.2** (Generalization of Concentration of Half Spaces). Consider the metric probability space, $(\mathcal{X}, \mu, \|\cdot\|_p)$, where $\mathcal{X} \subseteq \mathbb{R}^n$ and $p \geq 1$. Let $\{\boldsymbol{x}_i\}_{i \in [m]}$ be a set of $m$ instances sampled from $\mu$, and let $\widehat{\mu}_m$ be the corresponding empirical measure. Define the concentration functions regarding the collection of half spaces $\mathcal{HS}(n)$ with respect to $\mu$ as:

$$h_\mu^{(\ell_p)}\big(\alpha, \epsilon, \mathcal{HS}(n)\big) = \underset{\mathcal{E} \in \mathcal{HS}(n)}{\inf} \big\{\mu\big(\mathcal{E}_\epsilon^{(\ell_p)}\big) : \mu(\mathcal{E}) \geq \alpha\big\},$$

and let $h_{\widehat{\mu}_m}^{(\ell_p)}(\alpha, \epsilon, \mathcal{HS}(n))$ be its empirical counterpart with respect to $\widehat{\mu}_m$. For any $\delta \in (0, 1)$, there exists constants $c_0$ and $c_1$ such that with probability at least $1 - c_0 \cdot e^{-n \log n}$,

$$h_{\widehat{\mu}_m}^{(\ell_p)}\big(\alpha - \delta, \epsilon, \mathcal{HS}(n)\big) - \delta \leq h_\mu^{(\ell_p)}\big(\alpha, \epsilon, \mathcal{HS}(n)\big) \leq h_{\widehat{\mu}_m}^{(\ell_p)}\big(\alpha + \delta, \epsilon, \mathcal{HS}(n)\big) + \delta$$

holds, provided that the sample size $m \geq c_1 \cdot n \log n/\delta^2$.

**Remark 4.3.** Theorem 4.2 suggests that for the concentration of measure problem with respect to half spaces, in order to achieve $\delta$ estimation error with high probability, it requires $\Omega(n \log(n)/\delta^2)$ number of samples. Compared with Mahloujifar et al. (2019b), our method using half spaces requires fewer samples in theory to achieve the same estimation error.[3] For standard Gaussian

---

[3]The proposed estimators for $\ell_\infty$ and $\ell_2$ in Mahloujifar et al. (2019b) require $\Omega(nT \log(n) \log(T)/\delta^2)$ samples to achieve $\delta$ approximation, where $T$ is a predefined number of hyperrectangles or balls.

inputs, the empirical concentration with respect to half spaces is guaranteed to converge to the actual concentration as in (2.1), i.e., $\lim_{m \to \infty} h_{\widehat{\mu}_m}^{(\ell_p)}(\alpha, \epsilon, \mathcal{HS}(n)) = h_\mu^{(\ell_p)}(\alpha, \epsilon)$; whereas for distributions that are not Gaussian, there might exist a gap. However, this gap of empirical and actual concentration is shown to be uniformly small across various data distributions, as will be discussed in Section 6.2.

Based on Lemma 4.1, estimating the empirical concentration using half spaces as defined in (4.2) is equivalent to solving the following constrained optimization problem:

$$\begin{aligned}
\underset{\boldsymbol{w} \in \mathbb{R}^n, b \in \mathbb{R}}{\text{minimize}} \quad & \sum_{i \in [m]} \mathbb{1}\{\boldsymbol{w}^\top \boldsymbol{x}_i + b \leq \epsilon \|\boldsymbol{w}\|_q\} \\
\text{subject to} \quad & \frac{1}{m} \sum_{i \in [m]} \mathbb{1}\{\boldsymbol{w}^\top \boldsymbol{x}_i + b \leq 0\} \geq \alpha \text{ and } \|\boldsymbol{w}\|_2 = 1.
\end{aligned} \tag{4.3}$$

The optimal solution to (4.3) would be a half space $\mathcal{H}_{\boldsymbol{w}, b}$ that satisfies the following two properties: (1) approximately $\alpha$-fraction of data is covered by $\mathcal{H}_{\boldsymbol{w}, b}$, and (2) most of the remaining data points are at least $\epsilon$-away from $\mathcal{H}_{\boldsymbol{w}, b}$ under $\ell_p$-norm distance metric.

Note that we can always set $b$ to be the $\alpha$-quantile of the projections $\{-\boldsymbol{w}^\top \boldsymbol{x}_i : i \in [m]\}$ to satisfy the first property. In addition, to satisfy the second condition, inspired by the special case optimality results in Remark 3.4, we propose to search for a weight vector $\boldsymbol{w}$ such that both the $\ell_q$-norm of $\boldsymbol{w}$ is small and the variation of the given sample set along the direction of $\boldsymbol{w}$ is large. These searching criteria guarantee that the given dataset $\{\boldsymbol{x}_i\}_{i \in [m]}$, when projected onto $\boldsymbol{w}$ then normalized by $\|\boldsymbol{w}\|_q$, will have a large variance, which implies the second property.

We propose a heuristic algorithm to search for the desirable half space according to the aforementioned criteria (for the pseudocode and a detailed analysis of the algorithm, see Appendix C). It first conducts a principal component analysis with respect to the given empirical dataset, then iterates through all the principal components raised to an arbitrary power with normalization as candidate choices of the weight vector $\boldsymbol{w}$. Finally, based on the optimal choice of $b$, the algorithm outputs the best $\mathcal{H}_{\boldsymbol{w}, b}$ that achieves the smallest empirical measure with respect to the $\epsilon$-expansion of $\mathcal{H}_{\boldsymbol{w}, b}$. As we see in the experiments in the next section, this algorithm is able to find near-optimal solutions to the concentration problem for various datasets.

## 5 ERROR ANALYSIS

The goal of measuring concentration is to provide an empirical estimate of concentration of measure that minimizes the overall approximation error. Here, we describe the error components based on the general empirical framework for measuring concentration, and then discuss its implications on how to choose the collection of subsets $\mathcal{G}$ for the empirical concentration problem (4.1).

**Error Decomposition.** Let $(\mathcal{X}, \mu, \Delta)$ be the underlying input metric probability space and $\mathcal{G}$ be the selected collection of subsets for the empirical concentration problem. Suppose an algorithm that aims to solve the empirical concentration problem (4.1) returns $\mathcal{E}$ as a solution. For any $\alpha \in (0, 1)$ and $\epsilon \geq 0$, the approximation error between the empirical estimate of concentration and the actual concentration can be decomposed into three error terms:

$$\underbrace{h_\mu^{(\Delta)}(\alpha, \epsilon) - \widehat{\mu}_m(\mathcal{E}_\epsilon^{(\Delta)})}_{\text{approximation error}} = \underbrace{h_\mu^{(\Delta)}(\alpha, \epsilon) - h_\mu^{(\Delta)}(\alpha, \epsilon, \mathcal{G})}_{\text{modeling error}} + \underbrace{h_\mu^{(\Delta)}(\alpha, \epsilon, \mathcal{G}) - h_{\widehat{\mu}_m}^{(\Delta)}(\alpha, \epsilon, \mathcal{G})}_{\text{finite sample estimation error}}$$

$$+ \underbrace{h_{\widehat{\mu}_m}^{(\Delta)}(\alpha, \epsilon, \mathcal{G}) - \widehat{\mu}_m(\mathcal{E}_\epsilon^{(\Delta)})}_{\text{optimization error}}. \tag{5.1}$$

The *modeling error* denotes the difference between the actual concentration function and the concentration function with respect to the selected collection of subsets $\mathcal{G}$; the *finite sample estimation error* represents the generalization gap between the empirical concentration function and its limit; and the *optimization error* captures how well the algorithm approximates the empirical concentration problem. Such an error decomposition applies to both the empirical method proposed in this work as well as Mahloujifar et al. (2019b)'s, despite the different choices of $\mathcal{G}$.

The complexity of $\mathcal{G}$ and the complexity of its $\epsilon$-expansion $\mathcal{G}_\epsilon = \{\mathcal{E}_\epsilon : \mathcal{E} \in \mathcal{G}\}$ control the finite sample estimation error. So, $\mathcal{G}$ should be selected such that the empirical concentration function $h_{\hat{\mu}_m}^{(\Delta)}(\alpha, \epsilon, \mathcal{G})$ generalizes. If either $\mathcal{G}$ or $\mathcal{G}_\epsilon$ is too complex (e.g., it has unbounded VC-dimension), it will be difficult to control the generalization of the empirical concentration function (see Remark 3.4 in Mahloujifar et al. (2019b) for a detailed discussion).

There exist tradeoffs among the three error terms in (5.1), and it is unlikely that there is a uniformly good choice for $\mathcal{G}$ that minimizes all these error terms. In particular, increasing the complexity of $\mathcal{G}$ typically reduces the modeling error, since the feasible set of the concentration function with respect to $\mathcal{G}$ becomes larger. However, according to the generalization of concentration, this will also increase the finite sample estimation error. Therefore, we should consider the effect of all these error terms when choosing $\mathcal{G}$, including the hardness of the optimization problem with respect to the empirical concentration. It is favorable that the distance to any set in $\mathcal{G}$ has a closed-form solution, which enables exactly computing the empirical measure of the $\epsilon$-expansion of any set in $\mathcal{G}$. In addition, it will be easier to control the optimization error (i.e., develop an algorithm that produces tight estimates), if the empirical concentration problem is simpler. For instance, solving the empirical concentration problem with respect to the set of half spaces should be easier than solving it based on the union of hyperrectangles or balls, since there are more hyperparameters to optimize for the latter problem. Such simplicity further contributes to a tighter empirical estimate produced by our algorithm for the underlying concentration of measure problem.

Depending on the underlying distance metric Mahloujifar et al. (2019b) set $\mathcal{G}$ as a collection of the union of complement of hyperrectangles or the union of balls, whereas we choose $\mathcal{G}$ as the set of half spaces for any $\ell_p$-norm distance. In this work, we show that the set of half spaces is a superior choice of $\mathcal{G}$ for measuring the concentration of typical image benchmarks. Other choices of $\mathcal{G}$ may be preferred for different settings, but the same error decomposition and criteria will apply.

## 6 EXPERIMENTS

In this section, we evaluate our empirical method for estimating concentration under $\ell_\infty$-norm distance and comparing its performance to that of the method proposed by Mahloujifar et al. (2019b). We first demonstrate that the estimate produced by our algorithm is very close to the actual concentration for a spherical Gaussian distribution, and that our method is able to find much tighter bounds on the best possible adversarial risk for several image benchmarks. We then compare the convergence rates, and show that our method converges with substantially less data. Note that while we only provide results for the most widely-used $\ell_\infty$-norm perturbation metric adopted in the existing adversarial examples literature, our algorithm and experiments can be applied to any other $\ell_p$-norm.

### 6.1 ESTIMATION ACCURACY

First, we evaluate the performance of our algorithm under $\ell_\infty$-norm distance metric on a generated synthetic dataset consisting of 30,000 samples from $\mathcal{N}(\mathbf{0}, \mathbf{I}_{784})$. Since the proposed method follows from the analytical results of concentration of multivariate Gaussian distributions, we expect results produced by our empirical method to closely approach the analytical concentration on this simulated Gaussian dataset. We initially consider the case where $\epsilon = 1.0$ and $\alpha = 0.5$ for the actual concentration problem, requiring that the feasible set contains at least half of the data samples, and the adversary can perturb each entry by precisely the standard deviation of the underlying distribution. Our algorithm is able to produce a half space whose $\epsilon$-expansion has mean empirical measure $84.18\%$ over 5 repeated trials. According to Theorem 3.3 and Remark 3.4, the optimal value of the considered concentration problem is $84.13\%$. This implies that our method performs very well when the underlying distribution is Gaussian, while in stark contrast, the method by Mahloujifar et al. (2019b) is not able to find a region whose expansion has measure less than 1 on the same simulated set. In addition, we consider another setting for this dataset where $\epsilon = 1.0$ is set the same and $\alpha = 0.05$ is set to be much smaller. Similarly, we observe that our method significantly outperforms Mahloujifar et al. (2019b)'s in terms of the estimation accuracy (see Table 1 for the detailed comparison results).

Next, we evaluate our method on several image benchmarks. We set the values of $\alpha$ and $\epsilon$ to be the same as in Mahloujifar et al. (2019b) for the $\ell_\infty$ case. For example, we use $\alpha = 0.01$, $\epsilon \in$

Table 1: Comparisons between our method of estimating concentration with $\ell_\infty$-norm distance and the method proposed by Mahloujifar et al. (2019b) for different settings. For $\mathcal{N}(\mathbf{0}, \mathbf{I}_{784})$ with $\alpha = 0.5$ and $\epsilon = 1.0$, the previous method is unable to produce nontrivial estimate. Results for the previous method are taken directly from the original paper (except for the Gaussian results).

| Dataset | $\alpha$ | $\epsilon$ | Test Risk (%) | | Test Adv. Risk (%) | |
|---|---|---|---|---|---|---|
| | | | Prev. Method | Our Method | Prev. Method | Our Method |
| $\mathcal{N}(\mathbf{0}, \mathbf{I}_{784})$ | 0.05 | 1.0 | $6.21 \pm 0.44$ | $5.20 \pm 0.16$ | $89.75 \pm 0.80$ | $26.37 \pm 0.17$ |
| | 0.5 | 1.0 | - | $49.98 \pm 0.25$ | - | $84.18 \pm 0.11$ |
| MNIST | 0.01 | 0.1 | $1.23 \pm 0.12$ | $1.22 \pm 0.05$ | $3.64 \pm 0.30$ | $1.35 \pm 0.06$ |
| | | 0.2 | $1.11 \pm 0.10$ | $1.24 \pm 0.05$ | $5.89 \pm 0.44$ | $1.52 \pm 0.06$ |
| | | 0.3 | $1.15 \pm 0.13$ | $1.25 \pm 0.04$ | $7.24 \pm 0.38$ | $1.75 \pm 0.05$ |
| | | 0.4 | $1.21 \pm 0.09$ | $1.27 \pm 0.05$ | $9.92 \pm 0.60$ | $1.98 \pm 0.08$ |
| CIFAR-10 | 0.05 | 2/255 | $5.72 \pm 0.25$ | $5.14 \pm 0.13$ | $8.13 \pm 0.26$ | $5.28 \pm 0.12$ |
| | | 4/255 | $6.05 \pm 0.40$ | $5.22 \pm 0.20$ | $13.66 \pm 0.33$ | $5.68 \pm 0.21$ |
| | | 8/255 | $5.94 \pm 0.34$ | $5.22 \pm 0.16$ | $18.13 \pm 0.30$ | $6.28 \pm 0.13$ |
| | | 16/255 | $5.28 \pm 0.23$ | $5.19 \pm 0.08$ | $28.83 \pm 0.46$ | $7.34 \pm 0.15$ |
| Fashion-MNIST | 0.05 | 0.1 | $5.92 \pm 0.85$ | $5.33 \pm 0.14$ | $11.56 \pm 0.84$ | $6.04 \pm 0.13$ |
| | | 0.2 | $6.00 \pm 1.02$ | $5.34 \pm 0.14$ | $14.82 \pm 0.71$ | $6.82 \pm 0.19$ |
| | | 0.3 | $6.13 \pm 0.93$ | $5.24 \pm 0.10$ | $17.46 \pm 0.53$ | $8.01 \pm 0.19$ |
| SVHN | 0.05 | 0.01 | $8.83 \pm 0.30$ | $5.23 \pm 0.09$ | $10.17 \pm 0.29$ | $5.56 \pm 0.08$ |

$\{0.1, 0.2, 0.3, 0.4\}$ for MNIST, and $\alpha = 0.05$, $\epsilon \in \{2/255, 4/255, 8/255, 16/255\}$ for CIFAR-10. These $\alpha$ values were selected to roughly represent the standard error of the state-of-the-art classifiers.

Table 1 demonstrates the risk and adversarial risk with respect to the best produced subsets using both methods, computed on a separate test dataset. In our context of measuring concentration, risk refers to the empirical measure of the produced subset, while adversarial risk corresponds to the empirical measure of its $\epsilon$-expansion. We use a $50/50$ train-test split over the whole dataset to perform our evaluation, and determine the best exponent of each principal component based on a brute-force search. Though our method is deterministic for a given pair of training and testing sets, we account for the variance of our method over different train-test splits by repeating our experiments 5 times and reporting the mean and standard deviation of the results for each $(\alpha, \epsilon)$. It is worth noting that the randomness of the previous method is derived not only from the selection of the training and test sets, but also from the inherent randomness of the employed k-means algorithm.

We observe from Table 1 that in every case, the estimated adversarial risk is significantly lower for our method than for the one found by Mahloujifar et al. (2019b)'s. Since both methods restrict the search space to some special collection of subsets, these estimates can be viewed as valid empirical upper bounds of the actual concentration as defined in (2.1). Therefore, the fact that our results are significantly lower indicates that our algorithm is able to produce estimates that are much closer to the optimum of the targeted problem. In addition, when translated to adversarial robustness, these tighter estimates prove the existence of a rather robust classifier[4] that has risk at least $\alpha$, which further suggests that the underlying intrinsic robustness limit of each of these image benchmarks is actually much higher than previously thought.

For example, the best classifier produced by Mahloujifar et al. (2019b) has $18.1\%$ adversarial risk under $\ell_\infty$-perturbations bounded by $\epsilon = 8/255$ on CIFAR-10. However, our results demonstrate that the adversarial risk of the best possible robust classifier can be as low as $6.3\%$ given the same risk constraint, indicating the underlying intrinsic robustness to be above $93.7\%$. As the intrinsic robustness limits are shown to be very close to the trivial upper bound $1 - \alpha$ across all the settings, our results reveal that the concentration of measure phenomenon is not an important factor that causes the adversarial vulnerability of existing classifiers on these image benchmarks.

---

[4]Based on the ground-truth $f^*$ and the returned set $\mathcal{E}$ of our algorithm, this classifier can be simply constructed by setting $f(\boldsymbol{x}) = f^*(\boldsymbol{x})$ for $\boldsymbol{x} \notin \mathcal{E}$ and $f(\boldsymbol{x}) \neq f^*(\boldsymbol{x})$ for $\boldsymbol{x} \in \mathcal{E}$. Without knowing the ground-truth, we note that such classifier may or may not be learnable. The learnability of such $f$ is beyond the scope of this paper.

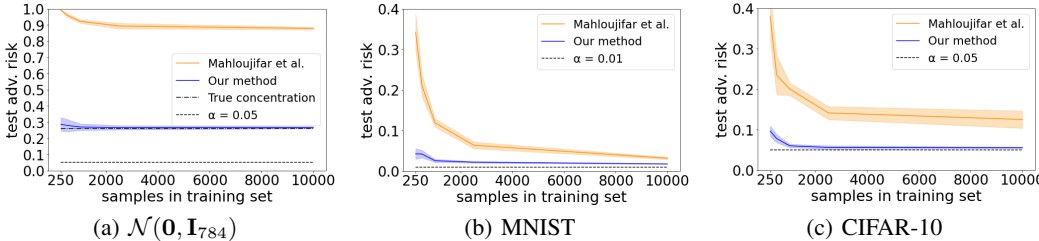

Figure 1: The convergence curves of the best possible adversarial risk estimated using our method and the method proposed by Mahloujifar et al. (2019b) as the number of training samples grows.

## 6.2 CONVERGENCE

Figure 6.2 shows the convergence rate of our method compared with that of the method proposed by Mahloujifar et al. (2019b) under $\ell_\infty$-distance for Gaussian data from $\mathcal{N}(\mathbf{0}, \mathbf{I}_{784})$ ($\alpha = 0.05$, $\epsilon = 1$), as well as for MNIST ($\alpha = 0.01$, $\epsilon = 0.1$) and CIFAR-10 ($\alpha = 0.05$, $\epsilon = 2/255$). We observe similar trends for other settings, thus we defer these results to Appendix D. For each graph, the horizontal $x$-axis represents the size of the dataset used to train the estimator, and the vertical $y$-axis shows the concentration bounds estimated for a separate test set, which is of size $30,000$ for each case. We generate the means and standard deviations for these convergence curves by repeating both methods 5 times for different randomly-selected training and test tests. For the proposed algorithm in Mahloujifar et al. (2019b), we tune the number of the hyperrectangles $T$ for the optimal performance based on the empirically-observed adversarial risk.

For the simulated Gaussian datasets, we include a horizontal line at $y = 0.2595$ to represent the true concentration of the underlying distribution, derived from Theorem 3.3 and Remark 3.4. This allows us to more accurately assess the convergence of our method, as it is the only case for which we know the optimal value that our empirical estimates should be converging to. We see that our estimates approach this line very quickly, coming within $0.01$ of the true value given only about 1,000 samples.

While we do not have such a theoretical limit for other datasets, the risk threshold $\alpha$ can be viewed as a lower bound of the actual concentration, which is useful in visually assessing the convergence performance of our method. We can see that for both MNIST and CIFAR-10, our estimates get very close to the horizontal line at $y = \alpha$ given several thousand training samples. Since the actual concentration must be no less than $\alpha$ and our estimated upper bound is approaching $\alpha$ from the above, we immediate infer that the actual concentration of these data distributions with $\ell_\infty$-norm distance should be a value sightly greater than $\alpha$. These results not only demonstrate the superiority of our method over the method of Mahloujifar et al. (2019b) in estimating concentration, but also show that concentration of measure is not the reason for our inability to find adversarially robust models for these image benchmarks.

## 7 CONCLUSION

Our results advance understanding of the intrinsic limits of adversarial robustness, strengthening the conclusion from Mahloujifar et al. (2019b) which asserted that concentration of measure is not the sole cause of the vulnerability of existing classifiers to adversarial attacks. We generalized the standard Gaussian Isoperimetric Inequality, and then leveraged theoretical insights from that to construct an efficient method for empirically estimating the concentration of arbitrary data distribution using samples. Our method is able to generalize to any $\ell_p$-norm distance metric, and surpasses previous approaches in both estimation accuracy and data-efficiency.

## AVAILABILITY

An implementation of our method, and code for reproducing our experiments, is available under an open source license from `https://github.com/jackbprescott/EMC_HalfSpaces`.

ACKNOWLEDGEMENTS

This work was partially funded by an award from the National Science Foundation SaTC program (Center for Trustworthy Machine Learning, #1804603). We thank Saeed Mauloujifar and Mohammad Mahmoody for valuable comments and discussions.

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

## A    PROOFS OF MAIN RESULTS IN SECTION 3

In this section, we provide the detailed proofs of our main theoretical results presented in Section 3.

### A.1    PROOF OF THEOREM 3.3

Before proving Theorem 3.3, we first lay out the following lemma regarding the monotonicity of $\ell_p$-norms. For a rigorous proof of this lemma, see Raıssouli & Jebril (2010).

**Lemma A.1** (Monotonicity of $\ell_p$). For any vector $\boldsymbol{x} \in \mathbb{R}^n$, the mapping $p \to \|\boldsymbol{x}\|_p$ is monotonically decreasing for any $p \geq 1$ (including $p = \infty$). That said, $\|\boldsymbol{x}\|_p \leq \|\boldsymbol{x}\|_q$ holds for any $p \geq q \geq 1$.

Now, we are ready to prove Theorem 3.3. In particular, we first include a high-level proof sketch, then present the complete proof after.

*Proof Sketch of Theorem 3.3.* We start with the spherical Gaussian distribution where $\nu = \gamma_n$. More specifically, we are going to prove that for any $\mathcal{E} \subseteq \mathbb{R}^n$ and $\eta \geq 0$,

$$\gamma_n\big(\mathcal{E}_\eta^{(\ell_p)}\big) \geq \Phi\big(\Phi^{-1}\big(\gamma_n(\mathcal{E})\big) + \eta\big) \text{ holds for } p \geq 2. \tag{A.1}$$

Note that for any vector $\boldsymbol{x} \in \mathbb{R}^n$, the mapping $p \to \|\boldsymbol{x}\|_p$ is monotonically decreasing for any $p \geq 1$, thus we can show that $\mathcal{E}_\eta^{(\ell_q)} \subseteq \mathcal{E}_\eta^{(\ell_p)}$ holds for any $p \geq q \geq 1$. Making use of the standard Gaussian Isoperimetric Inequality (Lemma 3.2), we then immediately obtain

$$\gamma_n\big(\mathcal{E}_\eta^{(\ell_p)}\big) \geq \gamma_n\big(\mathcal{E}_\eta^{(\ell_2)}\big) \geq \Phi\big(\Phi^{-1}\big(\gamma_n(\mathcal{E})\big) + \eta\big), \text{ for any } p \geq 2.$$

Moreover, to prove the concentration bound for general case where $\nu$ is the probability measure of $\mathcal{N}(\boldsymbol{\theta}, \boldsymbol{\Sigma})$, we build connections with the spherical Gaussian case by constructing a subset $\mathcal{A} = \{\boldsymbol{\Sigma}^{-1/2}(\boldsymbol{x} - \boldsymbol{\theta}) : \boldsymbol{x} \in \mathcal{E}\}$. Based on the affine transformation of Gaussian measure, we then prove:

$$\nu(\mathcal{E}) = \gamma_n(\mathcal{A}) \quad \text{and} \quad \nu(\mathcal{E}_\epsilon^{(\ell_p)}) \geq \gamma_n(\mathcal{A}_\eta^{(\ell_p)}), \quad \text{where } \eta = \epsilon/\|\boldsymbol{\Sigma}^{1/2}\|_p. \tag{A.2}$$

Finally, combining (A.1) and (A.2) completes the proof of Theorem 3.3. □

*Complete Proof of Theorem 3.3.* To begin with, we consider the special case where the underlying distribution is standard Gaussian ($\nu = \gamma_n$). Specifically, we are going to prove that for any $\mathcal{E} \subseteq \mathbb{R}^n$ and $\eta \geq 0$,

$$\gamma_n\big(\mathcal{E}_\eta^{(\ell_p)}\big) \geq \Phi\big(\Phi^{-1}\big(\gamma_n(\mathcal{E})\big) + \eta\big) \quad \text{holds for any } p \geq 2. \tag{A.3}$$

Let $p \geq q \geq 1$. According to the definition of $\epsilon$-expansion of a subset and Lemma A.1, we have

$$\mathcal{E}_\eta^{(\ell_q)} = \big\{\boldsymbol{x} \in \mathbb{R}^n : \exists\, \boldsymbol{x}' \in \mathcal{E} \text{ s.t. } \|\boldsymbol{x}' - \boldsymbol{x}\|_q \leq \eta\big\}$$
$$\subseteq \big\{\boldsymbol{x} \in \mathbb{R}^n : \exists\, \boldsymbol{x}' \in \mathcal{E} \text{ s.t. } \|\boldsymbol{x}' - \boldsymbol{x}\|_p \leq \eta\big\} = \mathcal{E}_\eta^{(\ell_p)} \tag{A.4}$$

where the inclusion is due to the fact that $\|\boldsymbol{x}' - \boldsymbol{x}\|_p \leq \|\boldsymbol{x}' - \boldsymbol{x}\|_q$ holds for any $\boldsymbol{x}'$ and $\boldsymbol{x}$. Therefore, by setting $q = 2$ in (A.4), we further obtain that for any $p \geq 2$,

$$\gamma_n\big(\mathcal{E}_\eta^{(\ell_p)}\big) \geq \gamma_n\big(\mathcal{E}_\eta^{(\ell_2)}\big) \geq \Phi\big(\Phi^{-1}\big(\gamma_n(\mathcal{E})\big) + \epsilon\big),$$

where the second inequality is due to the standard Gaussian Isoperimetric Inequality (Lemma 3.2). Thus, we have proven (A.3).

Now we turn to proving the concentration bound for the general Gaussian case. Let $\mathbf{U}\boldsymbol{\Lambda}\mathbf{U}^\top$ be the eigenvalue decomposition of $\boldsymbol{\Sigma}$, where $\mathbf{U} \in \mathbb{R}^{n \times n}$ is an orthonormal matrix and $\boldsymbol{\Lambda} \in \mathbb{R}^{n \times n}$ is a diagonal matrix consisting of all the eigenvalues. Since $\boldsymbol{\Sigma}$ is positive definite, the square root of $\boldsymbol{\Sigma}$ can be expressed as $\boldsymbol{\Sigma}^{1/2} = \mathbf{U}\boldsymbol{\Lambda}^{1/2}\mathbf{U}^\top$. Let $\boldsymbol{\Sigma}^{-1/2} = \mathbf{U}\boldsymbol{\Lambda}^{-1/2}\mathbf{U}^\top$ be the inverse matrix of $\boldsymbol{\Sigma}^{1/2}$.

Construct a subset $\mathcal{A}$ in $\mathbb{R}^n$ such that $\mathcal{A} = \{\boldsymbol{\Sigma}^{-1/2}(\boldsymbol{x} - \boldsymbol{\theta}) : \boldsymbol{x} \in \mathcal{E}\}$. Based on the construction of $\mathcal{A}$, we can then prove the following results for any $\mathcal{E} \subseteq \mathbb{R}^n$ and $\epsilon \geq 0$:

$$\nu(\mathcal{E}) = \gamma_n(\mathcal{A}) \quad \text{and} \quad \nu(\mathcal{E}_\epsilon^{(\ell_p)}) \geq \gamma_n(\mathcal{A}_\eta^{(\ell_p)}), \quad \text{where } \eta = \epsilon/\|\boldsymbol{\Sigma}^{1/2}\|_p. \tag{A.5}$$

First, we prove the equality $\nu(\mathcal{E}) = \gamma_n(\mathcal{A})$. Since $\nu$ is the probability measure of $\mathcal{N}(\boldsymbol{\theta}, \boldsymbol{\Sigma})$, we have

$$\nu(\mathcal{E}) = \Pr_{\boldsymbol{x} \sim \nu}[\boldsymbol{x} \in \mathcal{E}] = \Pr_{\boldsymbol{x} \sim \nu}[\boldsymbol{\Sigma}^{-1/2}(\boldsymbol{x} - \boldsymbol{\theta}) \in \mathcal{A}] = \Pr_{\boldsymbol{u} \sim \gamma_n}[\boldsymbol{u} \in \mathcal{A}] = \gamma_n(\mathcal{A}), \tag{A.6}$$

where the third inequality is due to the affine transformation of Gaussian random variables.

Next, we prove the remaining inequality in (A.5). By definition, for any $\boldsymbol{u}' \in \mathcal{A}_\eta^{(\ell_p)}$, there exists $\boldsymbol{u} \in \mathcal{A}$ such that $\|\boldsymbol{u}' - \boldsymbol{u}\|_p \leq \eta$. Let $\boldsymbol{x}' = \boldsymbol{\theta} + \boldsymbol{\Sigma}^{1/2}\boldsymbol{u}'$ and $\boldsymbol{x} = \boldsymbol{\theta} + \boldsymbol{\Sigma}^{1/2}\boldsymbol{u}$, then we have

$$\big\|\boldsymbol{x}' - \boldsymbol{x}\big\|_p = \big\|\boldsymbol{\Sigma}^{1/2}(\boldsymbol{u}' - \boldsymbol{u})\big\|_p \leq \|\boldsymbol{\Sigma}^{1/2}\|_p \cdot \|\boldsymbol{u}' - \boldsymbol{u}\|_p \leq \eta\|\boldsymbol{\Sigma}^{1/2}\|_p \leq \epsilon, \tag{A.7}$$

where the first inequality is due to the definition of induced matrix $p$-norm and the last inequality holds because $\eta = \epsilon/\|\boldsymbol{\Sigma}^{1/2}\|_p$. By the construction of $\mathcal{A}$ and the fact that $\boldsymbol{u} \in \mathcal{A}$, we have $\boldsymbol{x} \in \mathcal{E}$. Combining (A.7), this further implies that for any $\boldsymbol{u}' \in \mathcal{A}_\eta^{(\ell_p)}$, $\boldsymbol{\theta} + \boldsymbol{\Sigma}^{1/2}\boldsymbol{u}' \in \mathcal{E}_\epsilon^{(\ell_p)}$. Thus, we have

$$\nu\big(\mathcal{E}_\epsilon^{(\ell_p)}\big) \geq \nu\big(\boldsymbol{\theta} + \boldsymbol{\Sigma}^{1/2} \cdot \mathcal{A}_\eta^{(\ell_p)}\big) = \Pr_{\boldsymbol{x} \in \nu}\big[\boldsymbol{\Sigma}^{-1/2}(\boldsymbol{x} - \boldsymbol{\theta}) \in \mathcal{A}_\eta^{(\ell_p)}\big] = \gamma_n\big(\mathcal{A}_\eta^{(\ell_p)}\big), \tag{A.8}$$

where $\boldsymbol{\theta} + \boldsymbol{\Sigma}^{1/2} \cdot \mathcal{A}_\eta^{(\ell_p)}$ denotes the transformed subset $\{\boldsymbol{\theta} + \boldsymbol{\Sigma}^{1/2}\boldsymbol{u} : \boldsymbol{u} \in \mathcal{A}_\eta^{(\ell_p)}\}$. Therefore, based on (A.6) and (A.8), we prove the soundness of (A.5).

Finally, combining (A.3) and (A.5) completes the proof of Theorem 3.3. □

## A.2 PROOF OF THE OPTIMALITY RESULTS IN REMARK 3.4

*Proof.* First, we prove the optimality for the spherical Gaussian case, where $\nu = \gamma_n$ and $p > 2$. Let $\mathcal{H} = \mathcal{H}_{\boldsymbol{w},b}$ be a half space with axis-aligned weight vector, that said $\boldsymbol{w} = \boldsymbol{e}_j$ for some $j \in [n]$. Intuitively speaking, the $\epsilon$-expansion of $\mathcal{H}$ with respect to $\ell_p$-norm will only happen along the $j$-th dimension. More rigorously, we are going to prove the following results: for any $\epsilon \geq 0$,

$$\mathcal{H}_\epsilon^{(\ell_p)} = \mathcal{H}_\epsilon^{(\ell_2)} \quad \text{holds for any } p \geq 1. \tag{A.9}$$

By definition, $\mathcal{H} = \{\boldsymbol{x} \in \mathbb{R}^n : x_j + b \leq 0\}$. For any $\boldsymbol{x} \notin \mathcal{H}$, let $\widehat{\boldsymbol{x}} \in \mathcal{H}$ be the closest point of $\boldsymbol{x}$ in terms of $\ell_p$-norm. Since the weight vector $\mathbf{w}$ of $\mathcal{H}$ is axis-aligned, thus $\widehat{\boldsymbol{x}}$ will only differ from $\boldsymbol{x}$ by

the $j$-th element. That said, $\widehat{x}_{j'} = x_{j'}$ for any $j' \neq j$ and $\widehat{x}_j = -b$. Thus for any $p \geq 1$, we have $\|\boldsymbol{x} - \widehat{\boldsymbol{x}}\|_p = \|\boldsymbol{x} - \widehat{\boldsymbol{x}}\|_2 = x_j + b$. Based on this observation, we further obtain that for any $p \geq 1$,

$$\mathcal{H}_\epsilon^{(\ell_p)} = \{\boldsymbol{x} \in \mathbb{R}^n : x_j + b \leq \epsilon\} = \mathcal{H}_\epsilon^{(\ell_2)},$$

which proves (A.9). According to the Gaussian Isoperimetric Inequality (Lemma 3.2), we obtain

$$\gamma_n\big(\mathcal{H}_\epsilon^{(\ell_p)}\big) = \gamma_n\big(\mathcal{H}_\epsilon^{(\ell_2)}\big) = \Phi(\Phi^{-1}(\gamma_n(\mathcal{H})) + \epsilon).$$

Therefore, combining this with Theorem 3.3, we prove the optimality for the spherical Gaussian case.

Now we turn to prove the non-spherical Gaussian case with $p = 2$. Based on Theorem 3.3, the lower bound is $\Phi(\Phi^{-1}(\nu(\mathcal{E}) + \epsilon/\|\boldsymbol{\Sigma}^{1/2}\|_2)$ when $p = 2$. In the following, we are going to prove: if we choose $\mathcal{E} = \mathcal{H}_{\boldsymbol{v}_1, b}$, where $\boldsymbol{v}_1$ is the eigenvector with respect to the largest eigenvalue of $\boldsymbol{\Sigma}$, this lower bound is attained. Similarly to the proof of Theorem 3.3, we construct $\mathcal{A} = \{\boldsymbol{\Sigma}^{-1/2}(\boldsymbol{x} - \boldsymbol{\theta}) : \boldsymbol{x} \in \mathcal{E}\}$.

Note that when $\mathcal{E}$ is a half space, the constructed set $\mathcal{A}$ is also a half space. In particular, for the case where $\mathcal{E} = \mathcal{H}_{\boldsymbol{v}_1, b}$, for any $\boldsymbol{u} \in \mathcal{A}$, there exists an $\boldsymbol{x} \in \mathbb{R}^n$ such that $\boldsymbol{u} = \boldsymbol{\Sigma}^{-1/2}(\boldsymbol{x} - \boldsymbol{\theta})$ and $\boldsymbol{v}_1^\top \boldsymbol{x} + b \leq 0$. This implies that $\boldsymbol{v}_1^\top \boldsymbol{\Sigma}^{1/2} \boldsymbol{u} + \boldsymbol{v}_1^\top \boldsymbol{\theta} + b \leq 0$ for any $\boldsymbol{u} \in \mathcal{A}$. Since $\boldsymbol{v}_1$ is the eigenvector of $\boldsymbol{\Sigma}$, we further have that $\mathcal{A}$ is a half space with weight vector $\boldsymbol{\Sigma}^{1/2}\boldsymbol{v}_1 = \|\boldsymbol{\Sigma}^{1/2}\|_2 \cdot \boldsymbol{v}_1$.

Note that according to (A.2), as in the proof of Theorem 3.3, for any $\mathcal{E} \subseteq \mathbb{R}^n$, we have

$$\nu(\mathcal{E}) = \gamma_n(\mathcal{A}) \text{ and } \nu\big(\mathcal{E}_\epsilon^{(\ell_2)}\big) \geq \gamma_n\big(\mathcal{A}_\eta^{(\ell_2)}\big), \text{ where } \eta = \epsilon/\|\boldsymbol{\Sigma}^{1/2}\|_2.$$

For $\mathcal{E} = \mathcal{H}_{\boldsymbol{v}_1, b}$, based on the explicit formulation of $\ell_2$-distance to a half space, we can explicitly compute the $\eta$-expansion of $\mathcal{A}$ as

$$\mathcal{A}_\eta^{(\ell_2)} = \{\boldsymbol{u} \in \mathbb{R}^n : \boldsymbol{v}_1^\top \boldsymbol{\Sigma}^{1/2} \boldsymbol{u} + \boldsymbol{v}_1^\top \boldsymbol{\theta} + b \leq \eta \cdot \|\boldsymbol{\Sigma}^{1/2}\|_2\}.$$

When we set $\eta = \epsilon/\|\boldsymbol{\Sigma}^{1/2}\|_2$, it further implies that

$$\gamma_n\big(\mathcal{A}_\eta^{(\ell_2)}\big) = \Pr_{\boldsymbol{u} \sim \gamma_n}\big[\boldsymbol{v}_1^\top \boldsymbol{\Sigma}^{1/2} \boldsymbol{u} + \boldsymbol{v}_1^\top \boldsymbol{\theta} + b \leq \epsilon\big] = \Pr_{\boldsymbol{x} \sim \nu}\big[\boldsymbol{v}_1^\top \boldsymbol{x} + b \leq \epsilon\big] = \nu\big(\mathcal{E}_\epsilon^{(\ell_2)}\big).$$

Finally, according to the optimality of the standard Gaussian Isoperimetric Inequality (Lemma 3.2), this completes the proof. $\qquad\square$

# B  PROOFS OF THEORETICAL RESULTS IN SECTION 4

In this section, we present the proofs to the theoretical results presented in Section 4.

## B.1  PROOF OF LEMMA 4.1

*Proof of Lemma 4.1.* We only consider the case when $\boldsymbol{w}^\top \boldsymbol{x} + b > 0$, because $d_p(\boldsymbol{x}, \mathcal{H}_{\boldsymbol{w}, b})$ is zero trivially holds if $\boldsymbol{w}^\top \boldsymbol{x} + b \leq 0$. The problem of finding the $\ell_p$-distance from a given point $\boldsymbol{x}$ to a half space $\mathcal{H}_{\boldsymbol{w}, b}$ can be formulated as the following constrained optimization problem:

$$\min_{\boldsymbol{z} \in \mathbb{R}^n} \|\boldsymbol{z} - \boldsymbol{x}\|_p, \quad \text{subject to } \boldsymbol{w}^\top \boldsymbol{z} + b \leq 0. \tag{B.1}$$

Let $\widetilde{\boldsymbol{z}} = \boldsymbol{z} - \boldsymbol{x}$, then optimization problem (B.1) is equivalent to

$$\min_{\widetilde{\boldsymbol{z}} \in \mathbb{R}^n} \|\widetilde{\boldsymbol{z}}\|_p, \quad \text{subject to } \boldsymbol{w}^\top \widetilde{\boldsymbol{z}} + \boldsymbol{w}^\top \boldsymbol{x} + b \leq 0. \tag{B.2}$$

According to Hölder's Inequality, for any $\widetilde{\boldsymbol{z}} \in \mathbb{R}^n$ we have

$$-\|\boldsymbol{w}\|_q \cdot \|\widetilde{\boldsymbol{z}}\|_p \leq \boldsymbol{w}^\top \widetilde{\boldsymbol{z}} \leq \|\boldsymbol{w}\|_q \cdot \|\widetilde{\boldsymbol{z}}\|_p,$$

where $1/p + 1/q = 1$. Therefore, for any $\widetilde{\boldsymbol{z}}$ that satisfies the constraint of (B.2), we have

$$\boldsymbol{w}^\top \boldsymbol{x} + b \leq -\boldsymbol{w}^\top \widetilde{\boldsymbol{z}} \leq \|\boldsymbol{w}\|_q \cdot \|\widetilde{\boldsymbol{z}}\|_p. \tag{B.3}$$

Since $\|\boldsymbol{w}\|_2 = 1$, we have $\|\boldsymbol{w}\|_q > 0$, thus (B.3) further suggests $\|\widetilde{\boldsymbol{z}}\|_p \geq (\boldsymbol{w}^\top \boldsymbol{x} + b)/\|\boldsymbol{w}\|_q$.

Up till now, we have proven that the optimal value of (B.1) is lower bounded by $(\boldsymbol{w}^\top \boldsymbol{x} + b)/\|\boldsymbol{w}\|_q$. The remaining task is to show this lower bound can be achieved. To this end, we construct $\widehat{\boldsymbol{z}}$ as

$$\widehat{z}_j = x_j - \frac{\boldsymbol{w}^\top \boldsymbol{x} + b}{\|\boldsymbol{w}\|_q} \cdot \left( \frac{\boldsymbol{w}_j^q}{\sum_{j \in [n]} \boldsymbol{w}_j^q} \right)^{1/p}, \quad \text{for any } j \in [n],$$

where $1/p + 1/q = 1$. We remark that for the extreme case where $p = \infty$, such choice of $\widehat{\boldsymbol{z}}$ can be simplified as $\widehat{\boldsymbol{z}} = \boldsymbol{x} - (\boldsymbol{w}^\top \boldsymbol{x} + b) \cdot \text{sgn}(\boldsymbol{w})/\|\boldsymbol{w}\|_q$, where $\text{sgn}(\cdot)$ denotes the sign function for vectors. According to the construction, it can be verified that

$$\boldsymbol{w}^\top \widehat{\boldsymbol{z}} + b = (\boldsymbol{w}^\top \boldsymbol{x} + b) - \frac{\boldsymbol{w}^\top \boldsymbol{x} + b}{\|\boldsymbol{w}\|_q} \cdot \sum_{j \in [n]} w_j \cdot \left( \frac{\boldsymbol{w}_j^q}{\sum_{j \in [d]} \boldsymbol{w}_j^q} \right)^{1/p} = 0,$$

and $\|\widehat{\boldsymbol{z}} - \boldsymbol{x}\|_p = (\boldsymbol{w}^\top \boldsymbol{x} + b)/\|\boldsymbol{w}\|_q$. □

### B.2 PROOF OF THEOREM 4.2

*Proof of Theorem 4.2.* We write $\mathcal{HS}$ as $\mathcal{HS}(n)$ for simplicity. Let $S$ be a set of size $m$ sampled from $\mu$ and $\widehat{\mu}_S$ be the corresponding empirical measure. Note that the VC-dimension of $\mathcal{HS}(n)$ is $n + 1$ (see Mohri et al. (2018)), thus according to the VC inequality, we have

$$\Pr_{S \leftarrow \mu^m} \left[ \sup_{\mathcal{E} \in \mathcal{HS}(n)} \left| \widehat{\mu}_S(\mathcal{E}) - \mu(\mathcal{E}) \right| \geq \delta \right] \leq 8 e^{(n+1) \log(m+1) - m\delta^2/32}.$$

In addition, according to Lemma 4.1, the $\epsilon$-expansion of any half space is still a half space. Therefore, we can directly apply Theorem 3.3 in Mahloujifar et al. (2019b) to bound the generalization of concentration with respect to half spaces: for any $\delta \in (0, 1)$, we have

$$\Pr_{S \leftarrow \mu^m} \left[ h_{\widehat{\mu}_S}^{(\ell_p)}(\alpha - \delta, \epsilon, \mathcal{HS}) - \delta \leq h_\mu^{(\ell_p)}(\alpha, \epsilon, \mathcal{HS}) \leq h_{\widehat{\mu}_S}^{(\ell_p)}(\alpha + \delta, \epsilon, \mathcal{HS}) + \delta \right]$$

$$\geq 1 - 32 e^{(n+1) \log(m+1) - m\delta^2/32}.$$

Finally, assuming the sample size $m \geq c_0 \cdot n \log n / \delta^2$ for some constant $c_0$ large enough, then there exists positive constant $c_1$ such that

$$h_{\widehat{\mu}_S}^{(\ell_p)}(\alpha - \delta, \epsilon, \mathcal{HS}) - \delta \leq h_\mu^{(\ell_p)}(\alpha, \epsilon, \mathcal{HS}) \leq h_{\widehat{\mu}_S}^{(\ell_p)}(\alpha + \delta, \epsilon, \mathcal{HS}) + \delta$$

holds with probability at least $1 - c_1 \cdot e^{-n \log n}$. □

## C ALGORITHM FOR ESTIMATING CONCENTRATION

---

**Algorithm 1:** Heuristic Search for Robust Half Space under $\ell_p$-distance

---

**Input** : a set of samples $\{\boldsymbol{x}_i\}_{i \in [m]}$; strength $\epsilon$ (in $\ell_p$-norm); risk threshold $\alpha$; #iterations $S$.

$\mathbf{Q} \leftarrow$ compute the sample covariance matrix based on $\{\boldsymbol{x}_i\}_{i \in [m]}$;

$\mathcal{V} \leftarrow$ obtain the set of principal components by eigenvalue decomposition on $\mathbf{Q}$;

**for** $v \in \mathcal{V}$ **do**

    **for** $s = 1, 2, \ldots, S$ **do**

        $\boldsymbol{w} \leftarrow$ select from $\{\pm \text{pow}(\boldsymbol{v}, s)\}$;        // pow() is defined according to (C.1)

        $b \leftarrow \alpha$-quantile of the set $\{-\boldsymbol{w}^\top \boldsymbol{x}_i : i \in [m]\}$;

        $\text{AdvRisk}_\epsilon(\mathcal{H}_{\boldsymbol{w},b}) \leftarrow \sum_{i=1}^m \mathbb{1}(\boldsymbol{w}^\top \boldsymbol{x}_i + b \leq \epsilon \|\boldsymbol{w}\|_q)/m$;

    **end**

**end**

$(\widehat{\boldsymbol{w}}, \widehat{b}) \leftarrow \text{argmin}_{(\boldsymbol{w},b)} \text{AdvRisk}_\epsilon(\mathcal{H}_{\boldsymbol{w},b})$;

**Output :** $\mathcal{H}_{\widehat{\boldsymbol{w}}, \widehat{b}}$

---

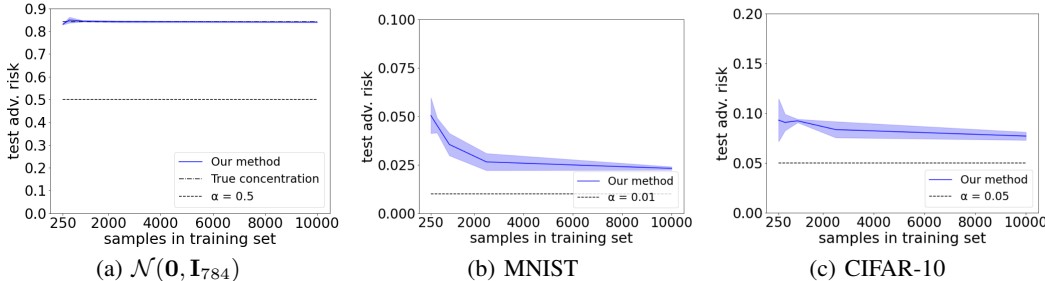

$$\text{(a) } \mathcal{N}(\mathbf{0}, \mathbf{I}_{784}) \qquad\qquad \text{(b) MNIST} \qquad\qquad \text{(c) CIFAR-10}$$

Figure 2: The convergence curves of the best possible adversarial risk estimated using our method under various settings as the sample size of the training dataset increases.

To solve the empirical concentration problem (4.3), Algorithm 1 searches for a desirable half space based on the principal components of the empirical dataset and their rotations defined by a power parameter. More specifically, the function $\mathrm{pow}()$ takes a vector $\boldsymbol{v} \in \mathbb{R}^n$ and a positive integer $s \in \mathbb{Z}^+$, and returns the normalized $s$-th power of $\boldsymbol{v}$ (with sign preserved):

$$\mathrm{pow}(\boldsymbol{v}, s) = \mathrm{sgn}(\boldsymbol{v}) \circ [\mathrm{abs}(\boldsymbol{v})]^s / \|\boldsymbol{v}^s\|_2 = \begin{cases} \boldsymbol{v}^s / \|\boldsymbol{v}^s\|_2, & \text{if } s \text{ is odd;} \\ \mathrm{sgn}(\boldsymbol{v}) \circ \boldsymbol{v}^s / \|\boldsymbol{v}^s\|_2, & \text{otherwise.} \end{cases} \tag{C.1}$$

Note that all the functions used in (C.1) are element-wise operations for vectors, where $\mathrm{sgn}(\boldsymbol{v})$, $\mathrm{abs}(\boldsymbol{v})$, $\boldsymbol{v}^s$ represent the sign, absolute value and the $s$-th power of $\boldsymbol{v}$ respectively, and the operator $\circ$ denotes the Hardamard product of two vectors.

Connected with the theoretical optimum regarding Gaussian spaces in Remark 3.4, the top principal component corresponds to the optimal choice of $\boldsymbol{w}$ if the perturbation metric is $\ell_2$-distance, whereas close-to-axis would be favourable for $\boldsymbol{w}$ when $p > 2$. In addition, as implied by the empirical concentration problem (4.3) and the monotonicity of $\ell_p$-mapping (Lemma A.1), the value of $\|\boldsymbol{w}\|_q$ will be more influential in affecting the $\epsilon$-expansion of half space as $p$ grows larger. For example, the $\ell_\infty$-norm of $\boldsymbol{w}$ can be as large as $\sqrt{n}$ for the worst case ($n$ denotes the input dimension), while $\|\boldsymbol{w}\|_\infty = 1$ if $\boldsymbol{w}$ aligns any axis. By searching through the region between each principal component and the closest axis, the proposed algorithm aims to find the optimal balance between $\|\boldsymbol{w}\|_q$ and the variance of the given data along $\boldsymbol{w}$ that leads to the smallest $\epsilon$-expansion. Although there is no theoretical guarantee that our algorithm will find the optimum to (4.3) for an arbitrary dataset, we empirically show (in Section 6) its efficacy in estimating concentration across various datasets.

Moreover, our algorithm is efficient in terms of both time and space complexities. Precomputing the principal components requires $O(mn^2 + n^3)$ time and $O(n^2)$ space to store them, where $m$ denotes the samples size and $n$ is the input dimension. For each iteration step, the time complexity of computing $\boldsymbol{w}, b$ and $\mathrm{AdvRisk}_\epsilon(\mathcal{H}_{\boldsymbol{w},b})$ is $O(mn)$, while the space complexity for saving the intermediate variables and the best parameters is $O(m + n)$. With $n$ outer iterations and $S$ inner iterations, the total time complexity is $O(n^3 + mn^2 S)$. The total space complexity is $O(n^2 + mn)$, where the extra $O(mn)$ denotes the initial space requirement for saving all the input data. For our experiments, we observe $\mathrm{AdvRisk}_\epsilon(\mathcal{H}_{\boldsymbol{w},b})$ is not sensitive to small increment of the exponent parameter $s$, thus we choose to increase $s$ in a more aggressive way, which further saves computation.

## D  ADDITIONAL EXPERIMENTS

Figure 2 shows the convergence performance of our algorithm under different experimental settings: $\alpha = 0.5$, $\epsilon = 1$ for the simulated Gaussian dataset, $\alpha = 0.01$, $\epsilon = 0.4$ for MNIST, and $\alpha = 0.05$, $\epsilon = 16/255$ for CIFAR-10. Under these additional settings, the algorithm proposed by Mahloujifar et al. (2019b) either cannot provide meaningful estimates of concentration, or takes a substantial amount of time to run. For instance, our algorithm takes around 2 days to generate the convergence curve on CIFAR-10 ($\alpha = 0.05$, $\epsilon = 16/255$), whereas the previous method is at least 5 times slower, due to the large number of rectangles $T$ needed. Thus, we only report the convergence curves of our method, where the standard deviations are calculated over 3 repeated trials.

