# OpenReview forum: "Improved Estimation of Concentration Under $\ell_p$-Norm Distance Metrics Using Half Spaces"
_ICLR.cc/2021/Conference — ICLR 2021 Poster_

### Official Review · AnonReviewer3 · 2020-10-28
**A good theory paper. Some parts of its contribution are more or less incremental, while some other parts look novel and promising.**

**Rating:** 6
**Confidence:** 4

**Review:**

$\{\\bf \{Problem\~ setting\~ and\~ summary\~ of\~ results\}\}$:

This paper gives a theoretical analysis on the fundamental limits of adversarial robustness under general $\\ell_{p}$-norm attacks. It basically builds upon a series of recent  works, mainly by Mahloujifar et al. (2019a and 2019b), which establish a fundamental relation between the  concentration function of input distribution, denoted by  $h^{\\left(\\ell_p\\right)}_{\\mu}\\left(\\epsilon,\\alpha\\right)$, and its intrinsic robustness, i.e., maximum possible adversarial robustness for any model class in a robust learning problem. The paper's main contribution is to propose a new way for approximating the empirical value of $h^\{\\left(\\ell_p\\right)\}_\{\\mu\}\\left(\\epsilon,\\alpha\\right)$ for a given input dataset of $m$ samples. Approximation is based on replacing the true input measure by its empirical version, and also limiting the error regions in Eq. (2.1) by a more manageable class (in this work: Half-spaces). In this regard, authors have investigated the two following issues: 1) consistency of the approximation and corresponding generalization bounds, and 2) tightness of the final bounds on intrinsic robustness.

$\{\\bf \{On\~the\~generalization\~aspect\}\}$: Authors have shown that estimating the concentration function $h^{\\left(\\ell_p\\right)}_{\\mu}\\left(\\epsilon,\\alpha\\right)$, for any $p\\in\\left(1,\\infty\\right\]$, through minimizing over the set of all half-spaces converges to its statistical value (i.e., when $m\\rightarrow\\infty$). The corresponding generalization bounds are given in Theorem 4.2, where a sample complexity similar to that of Mahloujifar et al., 2019, has been derived (authors claim that sample complexity has been improved w.r.t. prior works). Also, it should be noted that based on authors' claims in p.5, Mahloujifar et al. have only considered $\\ell_2$ and $\\ell_{\\infty}$-bounded attacks in their work while in this work $p$ is not restricted. Also, paper claims that the new proposed method converges to the optimal solution faster than previous works, however, no theoretical guarantees are given for this claim.

$\{\\bf \{On\~the\~tightness\~aspect\}\}$: Paper shows that when input data distribution is set to $\\mathcal{N}\\left(\\boldsymbol\{\\theta\},\\boldsymbol\{\\Sigma\}\\right)$, the concentration function becomes optimal and bounds are tight, if

a) $\\boldsymbol\{\\Sigma\}=\\boldsymbol\{I\}_n$ and $p>2$, or

b) $\\boldsymbol\{\\Sigma\}\\neq\\boldsymbol\{I\}_n$ and $p=2$.

For other cases, a novel and interesting bound is given (by extending the Gaussian Isoperimetric Inequality to $\\ell_p$ norms). However, achievability of bounds in these cases are not guaranteed in this paper.

------------------------------------------------------------------------------------------------------------------------------

$\{\\bf \{Comments\}\}$:

Paper is very well-motivated and fairly well-written. Mathematical notations are carefully crafted and results seem to be solid and understandable (I have not checked all of the proofs in Appendix). However, the contribution of the paper on the generalization aspect of the problem looks incremental and is similar to the prior works by Mahloujifar et al. (2019), however, in a relatively different setting. The main additions and improvements in this part are mostly heuristic and are backed solely through experiments.
The only exception is the sample complexity bound given in Theorem 4.2, where authors claim a substantial improvement has happened. However, this part need more explanation: How large parameter $T$ needs to be in (Mahloujifar et al., 2019) to guarantee the same error bound as this work? If $T$ is $O\\left(1\\right)$ w.r.t. dimension $n$, then I cannot see any considerable improvement (at least order-wise).

On the tightness issue, results are more novel and promising (in particular, Lemma 3.2). Again, the lack of theoretical guarantee for the achievability of Eq. (3.1) for the majority of cases has hampered the significance of the results in this part.

Overall, I believe this paper could potentially make a breakthrough if authors can provide further solid theoretical guarantees for any of the above issues. Still, I think the paper is publishable even in its current form.

---

> ### Author Response · Authors · 2020-11-18
> **A clarification on the choice of parameter T**
>
> We thank the reviewer for the feedback and thoughtful comments.
>
> > ... the sample complexity bound given in Theorem 4.2, where authors claim a substantial improvement has happened. However, this part need more explanation: How large parameter T needs to be in (Mahloujifar et al., 2019) to guarantee the same error bound as this work? If T is O(1) w.r.t. dimension n, then I cannot see any considerable improvement (at least order-wise).
>
> For the method proposed in Mahloujifar et. al. (2019b), the parameter $T$ captures the complexity of the selected collection of subset $\mathcal{G}$ as in Equation (4.1). Compared with Mahloujifar et. al. (2019b)'s, the asymptotic sample complexity improvement of our method is $O(T\log(T))$. This improvement, according to the generalization of concentration theorem, comes from the lower VC-dimension of the set of half spaces. When $T$ is chosen as a specific constant, we admit that this is not an order-wise improvement in theory. However, such improvement can still be large from an empirical perspective, especially when comparing methods given the same empirical dataset (the input dimension $n$ is fixed). In fact, the (optimal) value of $T$ can be rather large for certain experimental settings. For instance, Mahloujifar et. al. (2019b) sets $T=40$ as the optimal choice for the CIFAR-10 experiments under $\ell_\infty$ perturbations with $\epsilon=8/255$.
>
> Moreover, it is important to note that beyond the generalization aspect, the choice of $T$ (or more broadly, the choice of $\mathcal{G}$) also takes into account the optimization error and the modeling error with respect to the empirical estimation of concentration (see a detailed discussion of this in the new 'Error Analysis' Section). For each experimental setting, $T$ needs to be tuned based on the overall approximation error. Compared with the collection of union of $T$ hyperrectangles or balls (with optimally-tuned $T$), the collection of half-spaces is much simpler (with fewer hyperparameters to optimize for), which leads to a tighter estimate of the empirical concentration function than Mahloujifar et. al. (2019b)'s. Although we cannot prove theoretical guarantees for our proposed algorithm, its improved performance in terms of the overall approximation error is empirically observed for various datasets and settings as in Figure 1.

---

### Official Review · AnonReviewer4 · 2020-10-29
**Generalized Gaussian Concentration for Adversarial Robustness**

**Rating:** 6
**Confidence:** 4

**Review:**

Review
The authors generalized the Gaussian Isoperimetric Inequality to non-spherical Gaussian measures with $\ell_p$ metric structures $p\geq 2$. Building on the generalized inequality, they propose a sample-based algorithm to estimate the concentration of measure using half-spaces. The main contribution is Theorem 3.3 followed by the empirical sample based algorithm for half-spaces that requires $\Omega(n\log(n) /\delta^2)$ samples.


#### Reason for score
Overall, I vote for weak accept. The paper is very well written and it was a pleasure to go through it. Since I am not an expert in this research area, I have a low confidence in my decision. I would like to share a few minor concerns below to the authors that could improve further their submission.

#### Strong points:
- Paper is very well written and it was easy to go through the main contributions.
- A lot of effort is put in having a consistent,clear notation and formalism.

#### Minor Concerns
- I would expect that Theorem 3.3 to be already known in the literature, since Lemma 3.2 and Gaussian stability could probably derive such a result. At least, I believe that there might be tools that could make the proof of Theorem 3.3 shorter ( I had a quick look at it).
- Theorem 4.2: Why is this theorem stronger than the Mahloujifar et al. 2019? Both require $O(n\log(n) / \delta)$ samples but in footnote 3 you mention a parameter T. Is T a constant?
- I would suggest to move your algorithm from the appendix to the main text.
- Please make the argument stronger in the text on why "concentration of measure is not the main reason for the adversarial vulnerability of classifiers". Is this a definite conclusion?

#### Minor comments
- Abstract: Please rephrase the last sentence. "is not the main reason" could be rephrased.
- Sentence before "Contribution" : fix "to date to find"
- Why 8/255 in contribution? It looks like a magic number here.
- Notation: introduce square root of a matrix here.
- Notation: $\Delta$ mention that is a metric.
- Lemma 2.3 looks likea direct consequence of Defn. 2.1. Do you really need it?
- Defn. 3.1: do you need to define halfspaces? You can simply put it in text.
- All sets related should be measurable, i.e., Lemma 3.2, to avoid pathological cases.
- Theorem 4.2: is it true for $p\geq 1$ or $p\geq 2$?
- Footnote 3: Is T a constant or not? Please be specific. It is important since it compares with Theorem 4.2.

---

> ### Author Response · Authors · 2020-11-18
> **Response to Reviewer 4**
>
> We thank the reviewer for the feedback and constructive comments. For comments regarding the presentation of our paper, we have revised them accordingly based on your suggestions and highlighted the revision in blue in the updated paper.
>
> > I would expect that Theorem 3.3 to be already known in the literature, since Lemma 3.2 and Gaussian stability could probably derive such a result. At least, I believe that there might be tools that could make the proof of Theorem 3.3 shorter ( I had a quick look at it).
>
> When we started working in this direction, we expected this as well, since the GII is widely used, and it seems like it should have been previously generalized. However, to the best of our knowledge, we were unable to find such a generalization of the Gaussian Isoperimetric Inequality in any previous works.
>
> > Theorem 4.2: Why is this theorem stronger than the Mahloujifar et al. 2019? Both require $O(nlog(n))/\delta^2)$ samples but in footnote 3 you mention a parameter T. Is T a constant?
>
> Please refer to our response to Reviewer 3 for a detailed discussion regarding the choice of parameter $T$.
>
> > Please make the argument stronger in the text on why "concentration of measure is not the main reason for the adversarial vulnerability of classifiers". Is this a definite conclusion? The sentence "is not the main reason" in the abstract could be rephrased.
>
> We agree on the importance of rewording this to make our point clear. By saying "concentration of measure is not the main reason for the adversarial vulnerability of classifiers", we reach a conclusion contrasting the notion that some have argued (referenced earlier in the abstract) that concentration of measure in datasets is the fundamental cause of adversarial robustness. By using our novel techniques to show that these benchmark datasets are not concentrated, we rule out dataset intrinsic concentration (and equivalently, dataset intrinsic robustness) as a possible explanation for why no adversarially robust classifiers have been developed on these benchmark datasets. We revise this particular sentence of the abstract, which is reflected in the updated paper.
>
> > Why 8/255 in contribution? It looks like a magic number here.
>
> This is a common choice of attack strength for this setting, and the one we choose to highlight in contributions, but the results are similar across all of the settings in our experiments.
>
> > Lemma 2.3 looks like a direct consequence of Defn. 2.1. Do you really need it?
>
> We agree with the reviewer that Lemma 2.3 can be derived based on the definition of adversarial risk, but we still need Lemma 2.3 to connect the concentration of measure with the intrinsic robustness for a given robust classification problem. This explicitly characterizes our motivation for studying the concentration of measure problem.
>
> > Theorem 4.2: is it true for p >= 1 or p >= 2?
>
> Theorem 4.2 holds for any $p \geq 1$. It makes use of the fact that the $\epsilon$-expansion of the set of half-spaces with respect to any $\ell_p$-norm ($p\geq 1$) is still the set of half-spaces, which has VC-dimension $n+1$.

---

### Official Review · AnonReviewer1 · 2020-10-30
**Review1**

**Rating:** 7
**Confidence:** 3

**Review:**

Summary: The authors consider the problem of estimating intrinsic robustness using data samples. At a high level, intrinsic robustness is a measure that indicates the probability that a noisy version of a covariate would not be mislabeled. Mahloujifar et al. (2019a) have shown that estimating intrinsic robustness is closely related to the problem of estimating the concentration of measure. This problem focuses on finding a region such that it has the least likely epsilon neighborhood. In this work, the authors provide an efficient algorithm for measuring concentration empirically, which yields a more accurate estimation of the intrinsic robustness compared to the previous results.
One of the key technical components was generalizing the Gaussian Isoperimetric Inequality to non-spherical gaussian distributions (with distance metric being ell_p for p >2). This result leads to an optimal concentration result for special gaussian spaces.
Another utilized technique was using half spaces to estimate the intrinsic robustness. Basically, for estimating the intrinsic robustness, instead of solving an optimization problem over all possible regions, they only consider half-space regions. This approach yields a new formulation of the problem in the form of a linear program which can be solved efficiently.
In the end, the authors provide an empirical evaluation of their results.

Overall evaluation: The authors consider an important problem (understanding the robustness of a set of classifiers) from an interesting angle. The paper has substantial theoretical contributions that are empirically evaluated as well. The paper seems well-written.

Question: What is the main advantage of half-spaces (compared to hyperrectangles or balls), which results in a more efficient algorithm? Where does the improvement come from the generalization part (bounded VC-dimension of the extensions)? Or is it because using half-spaces, the problem can be nicely stated as an LP?

---

> ### Author Response · Authors · 2020-11-13
> **Advantages of using half-spaces**
>
> We would like to first thank the reviewer for the feedback and insightful comments.
>
> > What is the main advantage of half-spaces (compared to hyperrectangles or balls), which results in a more efficient algorithm?
>
> There are two key advantages of using half spaces specifically to define the error region:
>
>   1. Half spaces are simple and are easy to work with analytically, allowing us to prove Theorem 3.3 which demonstrates that for any Gaussian distribution, an analytically determinable half space must be the optimal solution to the problem. Though it is not strictly implied, this suggests that half spaces are likely to be close to optimal solutions in any distribution that is roughly Gaussian. Nothing we are aware of prohibits half spaces from also being good choices for highly non-Gaussian distributions, though in these cases the use of half spaces is only justified empirically. Our experimental results show that the use of half spaces is highly effective even on MNIST, which is non-Gaussian, significantly outperforming past approaches.
>
>   2. Also because of their simplicity, a near-optimal half space can be learned with less data than more complex regions like the union of hyperrectangles or the union of balls. This is the intuition behind why our new algorithm is so much more efficient than the approaches outlined by by Mahloujifar et al. (2019).
>
> > Where does the improvement come from the generalization part (bounded VC-dimension of the extensions)? Or is it because using half-spaces, the problem can be nicely stated as an LP?
>
> Compared with the union of hyperrectangles or balls, the improvement of using half-spaces comes from both of these two aspects. On the one hand, the VC-dimension of the set of half-spaces is lower, which implies a smaller finite sample estimation error (this is proved in Theorem 4.2 and summarized in Remark 4.3). This is the improvement that comes from the generalization part.
>
> On the other hand, half-spaces are easier to optimize for the empirical concentration problem as in Equation (4.1). Like you mentioned, it can be nicely stated as a simpler optimization problem as in Equation (4.3), where one only needs to determine the weight parameter $\mathbf{w}$ for a half-space (the intersection parameter $b$ can be easily selected based on $\alpha$). However, as for unions of hyperrectangles or balls, one needs to determine both the center location and the size for each hyperrectangle or ball, which is much more difficult from an optimization perspective. This simplicity of using half-spaces contributes to a tighter estimate for the corresponding empirical concentration function.

---

### Official Review · AnonReviewer2 · 2020-11-08
**Official Blind Review**

**Rating:** 7
**Confidence:** 2

**Review:**

Summary:
This paper considers the estimation of the concentration of measures, which possibly causes of the vulnerability of machine learning models to adversarial attacks. Towards such a goal, the authors first extend the Gaussian Isoperimetric Inequality to non-spherical Gaussian measures and arbitrary l_p norm. An algorithm that uses half-spaces as feasible set in the concentration of measure problem is then proposed. Empirical studies are conducted to show the efficiency as well as the efficacy of the proposed method when compared to [Mahloujifar et al. (2019b)].

Comment:
The paper is overall well written and the presentation is clear. While I am not an expert in this domain, I can easily follow the paper and understand the problem. While I cannot judge the novelty of Theorem 3.3, I very much appreciate Remark 3.4 which describes the limitations of Theorem 3.3 and clarifies the implication of such a result in several important special cases. In terms of the algorithm design, the choice of half spaces as feasible set in the concentration of measure problem seems natural given Lemma 3.2/Theorem 3.3 and is amenable to the generalization analysis.

---

### Author Response · Authors · 2020-11-18
**An update on our main paper**

Dear reviewers, we added a new section called "Error Analysis" to the main body of our paper (Section 5 in the updated version of our paper). This section elaborates our evaluation criteria when comparing different empirical methods for measuring concentration, and helps better explain the advantages of using the set of half-spaces for empirically measuring concentration.

---

### Decision · Program_Chairs · 2021-01-07
**Final Decision**

**Decision:**

Accept (Poster)

**Comment:**

High-quality theoretical paper that studies the connection between concentration of the data distribution and adversarial robustness. It contributes a method for more accurate estimation of concentration, which allows drawing stronger conclusions about adversarial robustness compared to previous work. The paper is highly technical, but written clearly and precisely. All reviewers give positive scores, with only minor negative comments.

One minor concern I have is that the potential audience of the paper might be small, given its highly technical nature and very specialized line of research it follows. Still, I believe it's a solid contribution, so I'm happy to recommend acceptance.